# Dual Benefits of Hydrogel Remediation of Cadmium-Contaminated Water or Soil and Promotion of Vegetable Growth under Cadmium Stress

**DOI:** 10.3390/plants12244115

**Published:** 2023-12-08

**Authors:** Jin Huang, Takehiko Gotoh, Satoshi Nakai, Akihiro Ueda

**Affiliations:** 1Graduate School of Advanced Science and Engineering, Hiroshima University, 1-4-1 Kagamiyama, Higashi-Hiroshima 739-8527, Hiroshima, Japan; d203238@hiroshima-u.ac.jp (J.H.); sn4247621@hiroshima-u.ac.jp (S.N.); 2Graduate School of Integrated Sciences for Life, Hiroshima University, 1-4-1 Kagamiyama, Higashi-Hiroshima 739-8527, Hiroshima, Japan

**Keywords:** hydrogel, heavy metal, cadmium immobilize, soil

## Abstract

This study aims to solve the problem of cadmium heavy metal ion pollution caused by the abuse of chemical fertilizers and activities such as mining, which pose a serious threat to the plant growth environment. We successfully synthesized DMAPAA (*N*-(3-(Dimethyl amino) propyl) acrylamide)/DMAPAAQ (*N*, *N*-Dimethyl amino propyl acrylamide, methyl chloride quaternary) hydrogels via free radical polymerization. Subsequently, we conducted experiments on this hydrogel for growing vegetables under cadmium stress conditions in aqueous solutions and soil. The cadmium capture capacity of DMAPAA/DMAPAAQ hydrogels under different cadmium ion concentrations and pH values was evaluated by using inductively coupled plasma optical emission spectrometry (ICP). The research results show that under the condition of pH = 7.3, the cadmium capture capacity of DMAPAA/DMAPAAQ hydrogels is the greatest. We used the Langmuir model to fit the adsorption data, and the correlation coefficient was as high as 0.96, indicating that the model fits well. The application of the hydrogels promoted the growth of vegetables in soil under cadmium stress conditions. The results showed that when the added amount of hydrogel was 4%, the dry weight of the vegetables was the largest. In addition, when the added amount of cadmium was 500 mg/kg and the added amount of hydrogel was 4%, the absorption of cadmium by the vegetables decreased to an undetectable level. In summary, the hydrogel successfully synthesized in this study can be effectively used to immobilize cadmium ions in soil while positively promoting the growth and yield of vegetables. This achievement has practical significance for solving the problem of heavy metal ion pollution.

## 1. Introduction

The current global situation of heavy metal contamination in soil has raised widespread concerns [1,2,3,4]. Different regions universally face varying degrees of heavy metal pollution, primarily caused by the release of heavy metal waste during industrial, mining, and agricultural processes. These pollutants enter the soil through processes such as atmospheric deposition and water leakage, causing the accumulation of heavy metals in the soil to exceed safety standards [5,6,7]. Heavy metal pollution has severe implications for the environment and human health. It not only hampers crop growth and yield but also permeates the food chain, ultimately negatively impacting human health and giving rise to a range of health issues. Consequently, there is an urgent need to seek efficient methods for addressing heavy metal pollution [8,9,10,11].

Currently, addressing soil pollution primarily relies on employing physical and chemical methodologies. Among these, soil displacement stands out as an effective means to significantly reduce the proportion of heavy metal content in contaminated soil [12]. On the other hand, thermal treatment techniques can be utilized to heat the polluted soil to extremely high temperatures, causing the metals to melt into a vitrified form, thereby mitigating their impact on the environment [13]. Additionally, the introduction of animal manure demonstrates efficacy in immobilizing heavy metal components in the soil, thereby diminishing their potential environmental hazards [14]. Meanwhile, the application of chemical agents that react with heavy metals to form precipitated compounds represents another potent remediation approach [15]. Moreover, bacteria exhibit promise by directly interacting with sites of heavy metal contamination through adherence to metal salts, resulting in metal dissolution [16]. However, these treatment technologies are not without their drawbacks, including relatively high costs and the potential generation of byproducts leading to secondary soil contamination issues.

Hydrogels are a type of polymer material with a three-dimensional network structure, exhibiting excellent adsorption properties, particularly in the removal of heavy metals. They employ surface functional groups such as carboxyl and hydroxyl moieties to engage in ion exchange with excess heavy metal ions in wastewater, securely immobilizing them within the hydrogel matrix [17,18]. Li et al.’s research team successfully synthesized a magnetic hydrogel featuring NH_2_ functional groups that, through coordination interactions, efficiently adsorb copper [19,20]. This coordination, facilitated by the formation of chemical bonds, effectively reduces the concentration of heavy metals in both aqueous solutions and soil environments [21]. Moreover, Zhang et al. engineered a dual-network hydrogel composed of alginate and polyvinyl alcohol. Relying on the active sites on the hydrogel surface with charges opposite to those of metal ions, they successfully removed heavy metals such as Pb (II) and Cr (VI) through electrostatic adsorption [22,23].

Liu et al. developed a composite material comprising FeS nanoparticles encapsulated within a lignin hydrogel matrix. The adsorption mechanism primarily involved the chemical reaction precipitation of cadmium sulfide, lignin complexation, hydrogel swelling, and nanoparticle adsorption [24]. Zhou et al. fabricated a novel composite hydrogel (LR-g-PAA/MMT/urea) that enhances soil fertility and mitigates the toxicity of heavy metals to organisms [25]. Dhiman et al. irrigated spinach with wastewater and incorporated water-absorbing gel–biochar to reduce the uptake of heavy metals by spinach [26]. Additionally, Zhou et al. devised a novel hydrogel that, upon pesticide release, undergoes the breaking of disulfide bonds on the hydrogel, generating thiol groups capable of complexing with heavy metals in the soil, thereby mitigating their toxicity to plants [27].

In this study, we synthesized DMAPAA/DMAPAAQ hydrogels using free radical polymerization and investigated their adsorption capabilities towards cadmium in aqueous environments. Furthermore, their introduction into soil was assessed to evaluate the growth of vegetables under cadmium stress. The aim of this research was to investigate the cadmium adsorption capacity of DMAPAA/DMAPAAQ hydrogels at varying pH levels, along with relevant thermodynamic and kinetic attributes. To simulate the adsorption process, we employed isotherm models such as Langmuir and Freundlich, while pseudo-first- and pseudo-second-order models were utilized to analyze the adsorption kinetics [28]. Additionally, the uptake of cadmium by vegetables in soil with the addition of DMAPAA/DMAPAAQ hydrogels under cadmium stress was evaluated, as was the content of various elements in vegetable leaves under the influence of both hydrogels and cadmium.

## 2. Results

### 2.1. Swelling Degree of Hydrogel

This study investigates the swelling behavior of hydrogels in both aqueous and soil environments. The results indicate that the addition of hydrogels significantly enhances the swelling capacity of both hydrogels and soil, whether placed in water or soil, as shown in Figure 1b. In aqueous conditions, the swelling capacity of hydrogels increased 18.6 times, while the incorporation of hydrogels into soil resulted in a 1.7-fold enhancement in the soil volumetric water content. This demonstrates the remarkable water-retention capabilities of hydrogels. The water-absorbing property of hydrogels is primarily attributed to the three-dimensional network structure formed by their polymer structure, which provides sufficient space for water molecules, leading to an expansion in the volume of the hydrogel. As shown in Figure 1a, the elemental analysis of the soil using EDX (EDX-7000 energy dispersive XRF spectrometers) revealed that the elements present were Si, Fe, Ca, Al, K, S, Ti, P, Mn, Zn, Sr, Cu, Rb, and Zr, with respective contents of 33.3%, 25.5%, 15.8%, 7.8%, 7.8%, 3.0%, 2.8%, 1.5%, 0.84%, 0.81%, 0.29%, 0.27%, 0.19.%, and 0.18%, some of which are beneficial for plant growth. Furthermore, when hydrogels are added to the soil, soil particles adhere to the surface of the hydrogel. The expansion of the hydrogel provides additional space for water in the soil, making it easier for plants to absorb water and acquire these crucial elements.

### 2.2. Effect of pH on Gel Adsorption

The aim of this study was to investigate the capture characteristics and effects of DMAPAA/DMAPAAQ hydrogels on cadmium ions under different pH conditions, as shown in Figure 2. The experimental results indicate that the capture capacity of the hydrogel for cadmium ions increases with increasing pH at pH 7, that the hydrogel exhibits the maximum capture capacity for cadmium, reaching 50 mg/g while at pH 2, and that the capture capacity is minimal, at only 5 mg/g. Under acidic conditions, the high concentration of H^+^ ions enhances the protonation degree of the hydrogel and increases its surface positive charge, resulting in repulsion, with the positive charge carried by cadmium ions. At this point, Cl^−^ forms stable complexes with Cd^2+^, which exists in the solution as ${CdCl}^{+}$, ${CdCl}_{2}^{0}$. When the pH is equal to 7, the tertiary amine in the hydrogel undergoes a protonation reaction in the aqueous solution, generating OH^−^ ions that form cadmium hydroxide precipitates, which are encapsulated within the hydrogel. This is the main mechanism by which the hydrogel effectively adsorbs the heavy metal cadmium.

### 2.3. Isothermal Adsorption

The experimental results in Figure 3 show that as the concentration of cadmium in the solution increases, the adsorption capture of the DMAPAA/DMAPAAQ hydrogels for cadmium ions also increases. In the Langmuir model, when pH = 7.3, the adsorption capacity of the hydrogel is 121 mg/g, significantly higher than that at pH = 5.7. This indicates that under acidic conditions, the adsorption of cadmium ions is mainly achieved through the Cl^−^ and Cd^2+^ from a stable complex. However, the adsorption capacity is limited. In neutral conditions, the amino groups on the hydrogel undergo protonation in water, and OH^−^ becomes the main adsorption group.

In the Freundlich model, Table 1 shows that the value of parameter n is between 1 and 10, indicating a spontaneous adsorption process. However, the correlation coefficient of the Langmuir model is higher than that of the Freundlich model at different pH values, indicating that the Langmuir model is more suitable for describing the adsorption process of the hydrogel on cadmium ions.

### 2.4. Adsorption Thermodynamic

By calculating the thermodynamic parameters, including the adsorption enthalpy ($∆H^{\theta }$), adsorption free energy ($∆G^{\theta }$), and adsorption entropy ($∆S^{\theta }$), the adsorption mechanism on the hydrogel was studied. Thermodynamic analysis was conducted on the adsorption experiments to elucidate the influence of different temperatures on the adsorption behavior of cadmium ions, as shown in Figure 4. It was found that when the temperature increased from 298 to 318 K, $∆G^{\theta }$ was consistently less than 0, indicating that the adsorption of cadmium ions by DMAPAA/DMAPAAQ hydrogels is spontaneous and feasible.

For the adsorption process of cadmium, $∆H^{\theta }$ is positive, and the adsorption capacity of cadmium ions increases with temperature, indicating that the adsorption of cadmium ions by the hydrogel is an endothermic reaction. At the same time, as shown in Table 2, $∆S^{\theta }$ is also positive, indicating an increase in disorder between the surface of the hydrogel and the interface with cadmium ions during the adsorption process. Thermodynamic studies help to understand whether the adsorption behavior of adsorbent materials is an exothermic or endothermic process, thereby expanding the practical applications of adsorbents.

### 2.5. Adsorption Kinetics

The experimental results in Figure 5a show that the entire adsorption process can be divided into two stages. In the initial stage, the adsorption rate is fast. After 200 min of adsorption, the DMAPAA/DMAPAAQ hydrogels’ capacity for absorbing cadmium has reached 90% of the equilibrium adsorption capacity. As time increases, the cadmium ion adsorption rate gradually decreases and basically reaches an equilibrium state after 24 h. This is because, in the initial stage, the amine groups on the hydrogel are protonated in water to form many OH^−^ ions, which combine with cadmium ions to form cadmium hydroxide precipitation. From Table 3, it can be observed that, at pH = 5.7, 7.3, the coefficients of the pseudo-first-order kinetic model for cadmium ion adsorption are higher than those of the second-order kinetic model. This shows that the hydrogel adsorption process is related to its concentration and that the adsorption sites mainly come from the amine groups on the hydrogel. The pseudo-first-order kinetic model is more suitable for the process of absorbing cadmium ions via hydrogels. Figure 5b shows the adsorption process of the ion internal diffusion model. The first stage is surface adsorption. At this stage, there are many active sites on the hydrogel. Under acidic conditions, adsorption mainly occurs through the chelation of chloride ions and cadmium ions on the hydrogel. Under neutral conditions, it is mainly the OH^−^ ions formed by the protonation of the amine group on the hydrogel that adsorb cadmium. The second stage is the internal diffusion adsorption stage of the hydrogel, which mainly occurs in the gaps of the hydrogel material. The adsorption rate at this stage decreases, and finally the adsorption equilibrium is reached.

### 2.6. Characterization of Hydrogel Structure

In this study, the structural changes in the DMAPAA/DMAPAAQ hydrogels after adsorption were thoroughly analyzed using infrared spectroscopy. Figure 6 reveals distinct absorption peaks in the adsorbed hydrogels: 1726 cm^−1^ corresponding to the stretching vibration peak of the amide C=O, 1514 cm^−1^ corresponding to the stretching vibration peak of C-N, 1311 cm^−1^ corresponding to the characteristic absorption peak of tertiary amine groups, and 3711 cm^−1^ corresponding to the characteristic peak of O-H. It is worth noting that the tertiary amines on the hydrogel undergo protonation in the aqueous solution, forming hydroxyl groups that bind with cadmium ions to form cadmium hydroxide precipitates, thereby enabling the hydrogel to encapsulate the cadmium. The hydrogel can be potentially applied in the form of tea bags for the removal of cadmium from soil through adsorption and precipitation without altering the soil’s acidity and alkalinity characteristics.

### 2.7. Soil Experiment

#### 2.7.1. Physical Picture of Vegetable Growth

This study recorded the growth status of plants using cameras, as shown in Figure 7. The results indicate that the addition of the hydrogel significantly improves the growth conditions of Swiss chard, presenting a larger growth status. This suggests that the application of the hydrogel has a positive promoting effect on plant growth under normal conditions and cadmium stress. This promoting effect has two main aspects: firstly, the hydrogel possesses a three-dimensional network structure. When added to the soil, it can absorb a large amount of water, keeping the soil moist, which is conducive to the reproduction of microorganisms and provides a more favorable growth environment. At the same time, the expansion of the hydrogel in the soil increases the gaps between soil particles, facilitating the respiration of roots and thus enhancing the growth rate of Swiss chard. Secondly, the DMAPAA/DMAPAAQ composite hydrogels could adsorb cadmium, reducing the biotoxicity of cadmium to vegetables and significantly improving the growth conditions of vegetables.

#### 2.7.2. Shoot Dry Weight

The dry weight of Swiss chard shoots was measured using an electronic balance, as shown in Figure 8. The results revealed that with an increase in the amount of hydrogel added, the dry weight of Swiss chard shoots also exhibited an increasing trend. Under low cadmium concentration conditions (less than 50 mg/kg), when the hydrogel addition was 4%, the maximum dry weight reached 0.765 g, which was 2.5 times higher compared to the situation without hydrogel addition. However, at a cadmium concentration of 500 mg/kg, with 4% hydrogel addition, the dry weight increased with the amount of hydrogel used, reaching a maximum value of approximately 0.503 g and a minimum value of 0.145 g. This indicates that the hydrogel not only possesses a capacity for cadmium adsorption but also enhances the moisture retention ability of soil, thereby increasing the vegetables’ yield. The dry weight of plants mainly comprises organic matter and trace elements. By assessing the dry weight of plants, we can accurately determine the content of elements such as carbon, hydrogen, and oxygen. This further contributes to a comprehensive evaluation of plant growth conditions and soil fertility.

#### 2.7.3. Cadmium Uptake in Swiss Chard

The cadmium content in Swiss chard was measured using ICP, as shown in Figure 9. As the amount of hydrogel added increased, the absorption of cadmium in Swiss chard gradually decreased. Under low cadmium concentration conditions (less than 50 mg/kg), the absorption of cadmium by Swiss chard was undetectable when no hydrogel was added, indicating that the soil had a certain fixation effect on cadmium. When the cadmium concentration increased to 500 mg/kg, the absorption of cadmium by Swiss chard increased significantly, but as the amount of hydrogel added increased, the absorption of cadmium by Swiss chard gradually decreased. Specifically, without the addition of the hydrogel, the absorption of cadmium by Swiss chard was 0.075 mg/g, while when the hydrogel addition was 4%, the absorption of cadmium by Swiss chard decreased to 0. This indicates that the addition of a hydrogel can effectively inhibit the absorption of heavy metal cadmium by Swiss chard. These results demonstrate that the application of hydrogels has a significant effect on reducing the absorption of cadmium in soil by vegetables and plays a positive role in reducing the phytotoxicity of heavy metal cadmium to plants.

#### 2.7.4. Other Elements

In this study, the content of elements in Swiss chard was analyzed utilizing inductively coupled plasma (ICP) measurement techniques. Among them, potassium (K), phosphorus (P), and magnesium (Mg) emerged as pivotal elements in the photosynthetic process, and it is possible that the chemical reactions involved in these elements increased the sugar content in the plant. The main elements in sugar are C, H, and O. Plants convert light energy into organic matter and store it in their bodies through photosynthesis. The introduction of hydrogels in this study holds significance regarding the exploration of their impact on plant photosynthesis. The application of hydrogels may enhance the efficiency of plant photosynthesis. This revelation holds substantial implications for future studies relating to the accumulation of organic compounds within plant organisms. In particular, there is broad applicative potential for using hydrogels in the cultivation of plants such as corn near mining areas. Furthermore, from Figure 10 it can found, in soil with a raised iron content (see Figure 1), plants exhibited a notable deficiency in iron absorption when the hydrogel was not incorporated, resulting in leaf chlorosis. Contrastingly, with the introduction of the hydrogel, there was a significant increase in the iron content within plant tissues. This indicates that the addition of hydrogels induces a shift in the utilization pattern of soil-bound iron, potentially by facilitating the adhesion of metallic ions to the surface of the hydrogel. Concurrently, plants release low-molecular-weight acids from their roots, promoting the reduction of trivalent iron to a more readily absorbable divalent state. Building upon this revelation, the co-cultivation of seaweed with a hydrogel and iron supplements emerges as a potential technique, given the substantial iron demands of seaweed. Additionally, elements such as Ca, Cu, and Zn stand out as essential elements, and Na is beneficial to some plants at a certain concentration. Monitoring the contents of these elements provides a means to assess the growth status and nutritional condition of plants, thereby providing a scientific basis for the optimization of cultivation environments.

## 3. Discussion

The use of hydrogels for the adsorption of heavy metals in water primarily relies on the functional groups on the surface of the hydrogel. Cadmium is a harmful heavy metal substance. Cadmium salts undergo hydrolysis in water, typically resulting in an acidic solution. In acidic solutions, the amine groups on the surface of the hydrogel sequester protons from the solution, increasing the positive charge on the hydrogel surface. Due to the same charges, this leads to hydrogel expansion [29,30,31]. At this point, the hydrogel primarily complexes and adsorbs cadmium with chloride ions. Under alkaline conditions, the hydrogel volume contracts, exhibiting pH responsiveness. This property finds widespread application in areas such as drug antimicrobial compounds and fertilizer release [32,33,34,35,36]. In neutral solutions, the tertiary amine group of DMAPAA is protonated in water to form OH^−^ ions. DMAPAAQ will exchange cadmium-containing anion compounds through ion exchange, such as nitrate ions and phosphate ions. These ions are useful nutrients for plants in the soil. Chloride ions are also released. Then, chloride ions and OH^−^, through the ion exchange method, let OH^−^ ions combine with the protonated tertiary amine group of DMAPAA. Finally, OH^−^ ions adsorb Cd ions to form a cadmium hydroxide precipitate, which is wrapped by the hydrogel. The adsorption mechanism is shown in Figure 11. This method exhibits good responsiveness to different pH conditions in the process of remediating the heavy metal cadmium. Additionally, this strategy holds broad applicative potential in areas such as drug antimicrobial compounds and fertilizer release. Introducing the hydrogel into soil, which is rich in elements like iron and manganese, leads to its expansion due to its water-absorbing properties. This, in turn, enlarges the gaps between soil particles, promoting oxidation reactions. During the oxidation process, the manganese and iron in the soil undergo oxidation reactions, forming manganese oxide (MnO_2_/Mn_2_O_3_) and iron oxide (Fe (OH)_3_/FeOOH) [37]. In this process, some heavy metal ions, such as cadmium, precipitate together with the oxides. Furthermore, the silicon and cadmium in the soil also undergo chelation, forming a Si–Cd structure to stabilize the cadmium ions [38]. The application of a hydrogel to the soil mainly enhances the plant yield via two processes: Firstly, plant growth relies on the process of photosynthesis, where chlorophyll absorbs sunlight and converts it into chemical energy through a series of chemical reactions, storing it in the form of biomass. Introducing a hydrogel into the soil forms a colloidal structure, widening the gaps between soil particles and enhancing the plant’s ability to absorb water. Additionally, hydrogels possess certain water-retaining properties, gradually releasing water to provide plants with more moisture and nutrients. Secondly, under neutral conditions, hydrogels exhibit a certain capacity for adsorbing heavy metals. Furthermore, they do not alter the pH value of the soil, effectively preventing soil acidification. Moreover, considering that soil is a highly complex environment with a diverse range of microbial species, the water-retaining property of hydrogels provides favorable conditions for microbial activity. For instance, microorganisms like *Rhizobium pusense* and *Bacillus* spp. can not only dissolve metals but can also participate in the oxidation and reduction processes of metals [39]. In summary, the application of hydrogels to soil plays a crucial role in promoting plant growth and increasing yield. Additionally, it provides favorable conditions for the environmental protection and microbial activity of soil [40].

## 4. Experimental Methods

### 4.1. Materials

The monomer DMAPAA/DMAPAAQ was obtained from KJ Chemicals Corporation (Tokyo, Japan), and *N*, *N*-dimethyl ethylenediamine (TEMED) was purchased from Nacalai Tesque Co., Ltd. (Kyoto, Japan). *N*, *N*′-methylenebisacrylamide (MBAA), ammonium persulfate (APS), and CdN_2_O_8_4H_2_O, CdCl_2_2.5H_2_O were acquired from Sigma-Aldrich, Inc. (St. Louis, MO, USA). In addition, the 1 mol·L^−1^ HCl and NaOH solution was purchased from Sigma-Aldrich Japan (Tokyo, Japan). The soil was purchased from NAFCO (Fukuoka, Japan). All the reagents used were of analytical grade and employed as received. The distilled water used in the experiments was produced in the laboratory.

### 4.2. Synthesis of Hydrogel

To synthesize the hydrogel, 1.315 g of DMAPAA and 2.584 g of DMAPAAQ monomers were weighed along with 0.193 g of MBAA crosslinker and 0.058 g of TEMED accelerator; this was added to a 20 mL volumetric flask (see Table 4). Distilled water was added to the flask, and the resulting mixture was stirred using a magnetic stirrer. In a separate 5 mL volumetric flask, a solution of 0.114 g of APS initiator was prepared using distilled water. Both the monomer solution and initiator solution were purged with nitrogen for 45 min to eliminate any oxygen and prevent the inhibition of free radicals. The initiator solution was then added to the monomer solution and stirred for 20 s. The resulting mixture was transferred to three plastic tubes using a pipette and immersed in an aqueous solution at 25 °C for 24 h. The resulting gels were removed from the tubes and cut into uniform cylindrical shapes, and then washed with methanol for 24 h using Soxhlet extractor (Asahi Glass plant Inc., Arao City, Japan) equipment to remove any unreacted components. The gels were subsequently air-dried at room temperature and further dried in a 50 °C oven for 24 h. Finally, the dried gels were ground into a powder using a grinder.

### 4.3. Swelling Properties of Hydrogels

#### 4.3.1. Swelling Degree of Hydrogel in Water Solutions

A hydrogel weighing 0.02 g (Md) was added to 40 mL of water solution. The mixtures were allowed to equilibrate at room temperature for 24 h to achieve swelling equilibrium. The swollen gels were subsequently filtered through filter paper and their weight recorded as Ms, calculated using the following formula:$$W_{a}=\frac{\left(Ms-Md\right)}{Md}$$

#### 4.3.2. Soil Water Holding Rate

Weighed soil of 10 g was combined with 0.1 g of hydrogel and recorded as *m*_1_. Subsequently, an appropriate amount of water was added to the beaker at room temperature, allowing the complete saturation of the mixture. The mass of the resulting mixture and water was measured via filtration using filter paper and recorded as *m*_2_, calculated using the following formula:$$W_{b}=\frac{\left(m_{2}-m_{1}\right)}{m_{1}}$$

### 4.4. Adsorption Experiment

#### 4.4.1. Adsorption Thermodynamic Experiments with Different pH Values

First, 1000 ppm of cadmium solution was diluted to a concentration of 5, 50, 100, 300, and 500 mg·L^−1^. Then, 25 mL of each diluted solution was transferred into 50 mL plastic centrifuge tubes. The pH of the solutions was adjusted to 7.3 by adding 1 mol·L^−1^ NaOH solution to a final volume of 40 mL. Furthermore, 20 mg of the adsorbent was introduced into each tube. The tubes were placed in a water-bath constant-temperature shaker at 25 °C, with continuous shaking at 150 rpm for 24 h. Following the adsorption period, the samples were filtered through a 0.22 μm membrane filter and analyzed using ICP.

#### 4.4.2. pH Effect Experiment

The process began with the dilution of a 1000 ppm cadmium solution to a concentration of 100 mg·L^−1^. Then, 25 mL of this diluted solution was carefully transferred into a 50 mL plastic centrifuge tube. To adjust the pH values to 2.0, 3.0, 4.0, 5.0, 6.0, and 7.0, 1 mol·L^−1^ of NaOH or HCl solution was added until the final volume reached 40 mL. Subsequently, 20 mg of the adsorbent material was introduced into the mixture. The entire system was placed in a water-bath constant-temperature shaker, where continuous adsorption was conducted at 25 °C and 150 rpm for a duration of 24 h. Following adsorption, the samples were filtered through a 0.22 μm membrane filter and analyzed using ICP.

#### 4.4.3. Adsorption Kinetic Experiment

The procedure began with the dilution of a 1000 ppm cadmium solution to achieve a concentration of 100 mg·L^−1^. Following this, 40 mL of the diluted solution was prepared, and the pH levels were adjusted to 5.7 and 7.3 using a 1 mol·L^−1^ NaOH solution. Next, 20 mg of the adsorbent material was introduced into the system. The entire setup was placed in a water-bath constant-temperature shaker, maintaining a constant temperature of 25 °C with a shaking rate of 150 rpm. The adsorption process was conducted for specific time intervals (12, 48, 72, 120, 180, 240, 300, 1200, 1320, and 1440 min). At each predetermined time point, samples were withdrawn from the system and subsequently filtered through a 0.22 μm membrane filter. The filtered samples were then analyzed using ICP.

#### 4.4.4. Adsorption Thermodynamics at Different Temperatures

The 1000 ppm original cadmium solutions were diluted to concentrations of 5, 50, 100, 300, and 500 mg·L^−1^, respectively. For each concentration, 25 mL of the diluted solution was transferred into a plastic centrifuge tube with a volume of 50 mL. Subsequently, the pH of each solution was adjusted to 5.7 and 7.3 using 1 mol·L^−1^ of NaOH solution to reach a final volume of 40 mL. Next, 20 mg of the adsorbent material was added to each solution. The prepared samples were placed in a water-bath constant-temperature shaker, maintaining constant temperatures of 25 °C and 35 °C, respectively, with a shaking rate of 150 rpm. The adsorption process was conducted continuously for 24 h. After the adsorption period, the samples were filtered through 0.22 μm membrane filters to remove any particulate matter. The filtrates were subsequently analyzed using ICP to measure the desired parameters.

#### 4.4.5. Adsorption Model

The isotherm adsorption of cadmium onto the DMAPAA/DMAPAAQ hydrogels at pH 5.7 and pH 7.3 is shown in Figure 3. The Langmuir and Freundlich models were employed to fit the experimental data. The Langmuir model assumes that the adsorption of the adsorbent to the target is only a single-layer adsorption and that the interaction force between the adsorbed molecules can be ignored. The relationship between the adsorption capacity (*q_e_*) and the cadmium concentration (*C_e_*) in the solution is described by the following equation:$$q_{e}=\frac{q_{max}K_{L}C_{e}}{1+K_{L}C_{e}}$$

Here, $q_{e}$ represents the adsorption capacity at equilibrium (mg·g^−1^), $q_{max}$ is the maximum adsorption capacity calculated by the Langmuir model (mg·g^−1^), and *K_L_* is the Langmuir constant (L·mg^−1^).

The Freundlich model assumes that the adsorption of the adsorbent to the target involves heterogeneous multilayer adsorption and that the relationship between the adsorption capacity $q_{e}$ and cadmium concentration in the solution (Ce) can be described by the following equation:$$q_{e}=K_{F}C_{e}^{1/n}$$

Here, $q_{e}$ denotes the adsorption capacity at equilibrium (mg·g^−1^), while *K_F_* and 1/*n* are the Freundlich model constants (mg·g^−1^).

To investigate the thermodynamic properties of the adsorption of cadmium by the DMAPAA/DMAPAAQ hydrogel at pH = 7.3, the calculations for Gibbs free energy, enthalpy change, and entropy change were performed using the following formulas:$$K_{D}=\frac{q_{e}}{C_{e}}$$
$$∆G^{\theta }=-RTInK_{D}$$
$$∆G^{\theta }=∆H^{\theta }-T∆S^{\theta }$$

In the equations, $K_{D}$ represents the distribution coefficient, *R* is the ideal gas constant (8.314 J mol^−1^ K^−1^), *T* is the thermodynamic temperature (K), $∆G^{\theta }$ represents the Gibbs free energy (kJ/mol), $∆H^{\theta }$ represents the enthalpy change (kJ/mol), and $∆S^{\theta }$ represents the entropy change (J mol^−1^ K^−1^).

The study of adsorption kinetics can reveal the adsorption rate and mechanism of adsorbent materials towards the target substances. In this experiment, the adsorption kinetics of the DMAPAA/DMAPAAQ hydrogels on cadmium were investigated, and two isothermal adsorption models, namely the pseudo-first-order kinetic model and the pseudo-second-order kinetic model, were employed for fitting. The pseudo-first-order kinetic model describes the adsorption process using the following equation:$$q_{t}=q_{e}(1-e^{-k_{1}t})$$

In the equations, $q_{t}$ represents the adsorption capacity at time t (mg/g), $q_{e}$ represents the adsorption capacity at equilibrium (mg/g), and $k_{1}$ is the rate constant of the pseudo-first-order kinetics (min^−1^). The pseudo-second-order kinetic model describes the adsorption process using the following formula:$$q_{t}=\frac{\left(K_{2}q_{e}^{2}t\right)}{1+K_{2}q_{e}t}$$

Among them, $q_{t}$, $q_{e}$, and *t* have the same meaning as the first-order kinetic model, and $k_{2}$ is the rate constant of the pseudo-second-order kinetics model (g·mg^−1^·min^−1^).

### 4.5. Experimental Design

A total of 800 g of wet soil was carefully weighed and transferred to a 1/10000a Neubauer pot. Subsequently, the DMAPAAA/DMAPAAQ hydrogels were added to the soil at concentrations of 0%, 2%, and 4% (*w*/*w*). CdCl_2_ solutions were added to the soil at concentrations of 0, 5, 50, and 500 mg/kg. The mixture was thoroughly mixed to ensure the uniform distribution of the cadmium and gel in the soil. The soil was then allowed to attain a semi-moist state over a few days and subsequently divided into two equal portions. Small holes were made in each of the four directions of the pot, and four seeds of Swiss chard (*Beta vulgaris* L. var. *cicla*) were placed in each hole.

#### 4.5.1. Dry Weight of Swiss Chard

After two months of cultivation in the greenhouse, eight Swiss chard plants were harvested by cutting the roots with scissors and washing the soil from the roots with water. Harvested plants were placed in envelope bags and dried in a drying oven at 70 °C for several days until completely dried. The dry weights of Swiss chard were measured using an electronic balance.

#### 4.5.2. Plant Digestion

Dried plants were crushed using a magnetic grinder (BMS-A20TP, Biomedical Science Corp., Tokyo, Japan) for 15 min at 700 rpm/min, after which approximately 50 mg of each sample was weighed and recorded. Subsequently, 2 mL of nitric acid was added to each of the centrifuge tubes, followed by 0.5 mL of hydrogen peroxide, and then subjected to digestion using a heat block incubator (DTU-2BN, Taitec Corp., Saitama, Japan). The digestion process began at 80 °C for 30 min, followed by a further increase in temperature to 120 °C for 2 h, and left overnight in a fume hood. After digestion, the resulting solution was filtered through filter paper and made up to 25 mL in a volumetric flask. Finally, the concentration of various elements in the vegetables was determined using ICP.

### 4.6. Characterization

The concentration of the heavy metal cadmium in the solution was measured using ICP (SPS-3500, Shimadzu Corp., Kyoto, Japan). The daily growth of the vegetable was monitored and recorded using a high-resolution camera (Xiaomi12 Technology Co., Ltd., Beijing, China). The elemental concentration of the vegetable was analyzed using ICP. The pH of the soil was measured using a precise pH meter (Hanna Groline Soil PH Tester HI981030, Rome, Italy). The soil properties were further characterized using EDX (EDX-7000, Shimadzu Corp., Kyoto, Japan) analysis. The structure of the gels was analyzed using IR (FTIR, IRTracer-100, Shimadzu Corp., Kyoto, Japan).

## 5. Conclusions

In this study, DMAPAA/DMAPAAQ hydrogels were successfully synthesized using the free radical polymerization method, and their adsorption performance was comprehensively evaluated under different pH values and cadmium concentrations. The results show that the hydrogels exhibited excellent cadmium capture performance under neutral conditions. We used the Langmuir model to fit the data and obtained a correlation coefficient as high as 0.97, indicating that the Langmuir model is more suitable for describing the process of capturing cadmium in hydrogels. Under neutral conditions, cadmium is wrapped around the hydrogel mainly through the precipitation of cadmium hydroxide formed by OH^−^, which is generated via the protonation of the tertiary amine on the hydrogel structure in water. We added the hydrogel to soil and observed the growth of Swiss chard under cadmium stress. Under low-concentration cadmium conditions (less than 50 mg/kg), when the added amount of hydrogel was 4%, the dry weight of Swiss chard reached up to 0.765 g, which is 2.5 times higher than when no hydrogel was added. When the cadmium concentration reached 500 mg/kg and the hydrogel addition amount was 4%, the dry weight of Swiss chard increased with the increase in the amount of hydrogel, reaching a maximum of 0.503 g and a minimum of 0.145 g. Under high-cadmium-concentration conditions (500 mg/kg), when no hydrogel was added, the cadmium absorbed by Swiss chard was 0.075 mg/g. After adding 4% hydrogel, the absorption of cadmium by Swiss chard dropped to an undetectable level. In summary, the results of this study show that in a cadmium-polluted environment, the use of DMAPAA/DMAPAAQ hydrogels can effectively promote the growth of vegetables and enhance the cadmium immobilization capacity of hydrogels in soil. To further enhance the soil adsorption capacity of cadmium, future research could consider the combined application of hydrogels with microorganisms or various wastes, such as steel slag, seaweed, animal manure, and plant ash.

## Figures and Tables

**Figure 1 plants-12-04115-f001:**
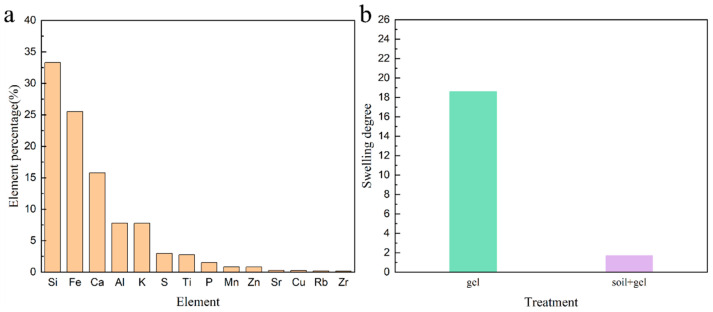
(**a**) Main elements contained in soil; (**b**) swelling degree of hydrogel in water and soil.

**Figure 2 plants-12-04115-f002:**
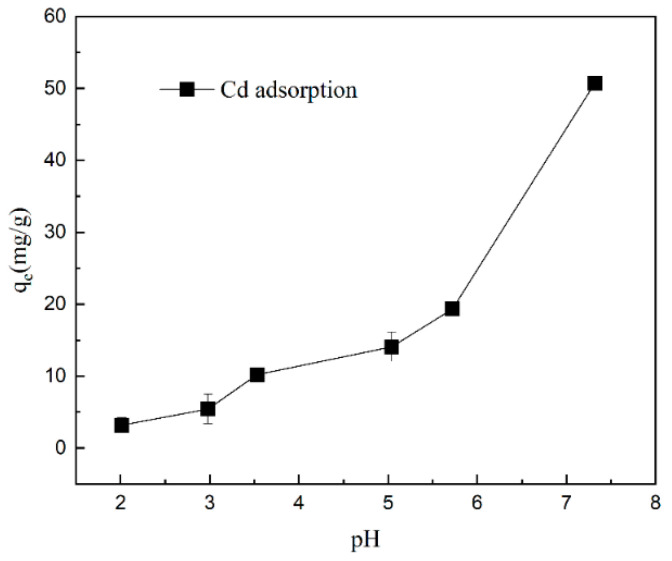
Effect of hydrogel adsorption capacity on cadmium ions at different pH values.

**Figure 3 plants-12-04115-f003:**
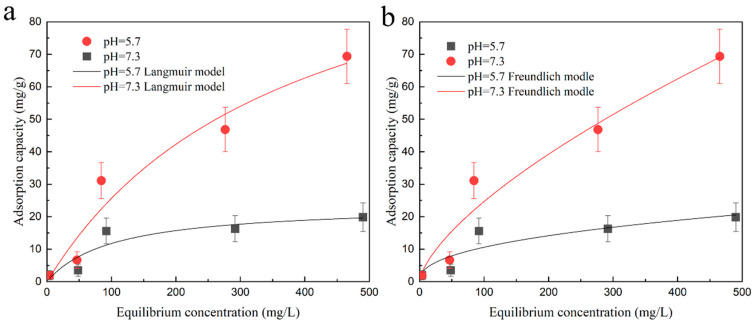
Hydrogel isothermal adsorption curve for cadmium ions at pH = 5.7, 7.3: (**a**) Langmuir simulation fitting curve; (**b**) Freundlich simulation fitting curve.

**Figure 4 plants-12-04115-f004:**
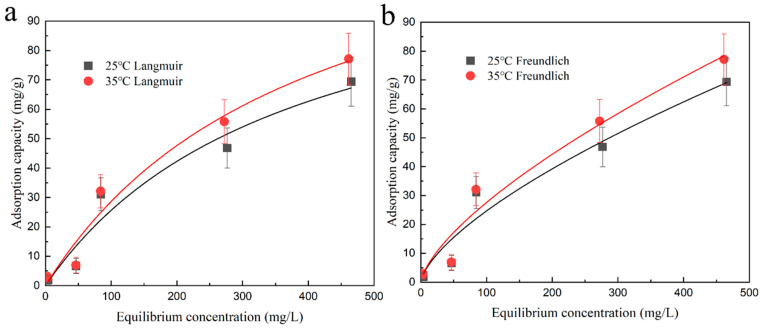
Adsorption isotherm fit curves of hydrogel at different temperatures: (**a**) Cd^2+^ adsorption Langmuir model simulated fit curve; (**b**) Cd^2+^ adsorption Freundlich model simulated fit curve.

**Figure 5 plants-12-04115-f005:**
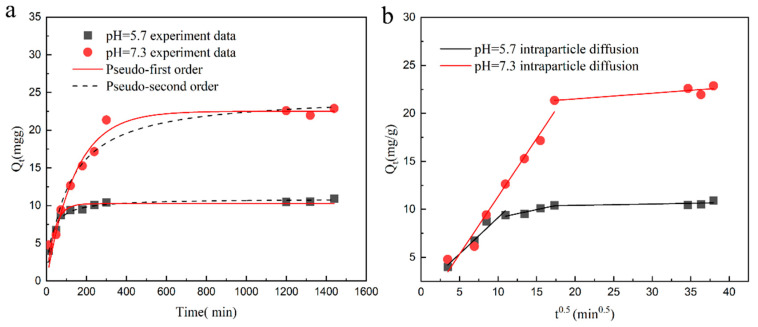
(**a**) Pseudo-first-order and pseudo-second-order dynamic model diagrams of cadmium adsorption by hydrogel; (**b**) hydrogel adsorption and diffusion model.

**Figure 6 plants-12-04115-f006:**
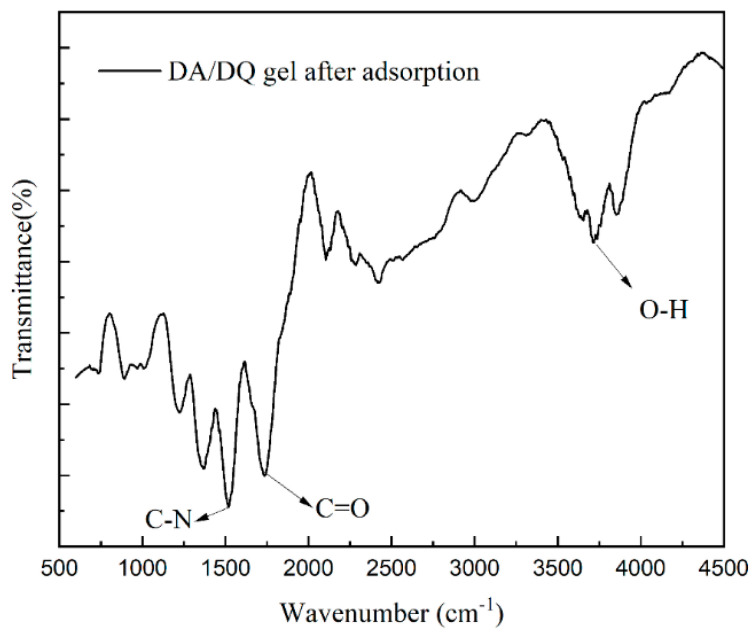
FTIR spectrum of hydrogel after adsorption at 600–4500.

**Figure 7 plants-12-04115-f007:**
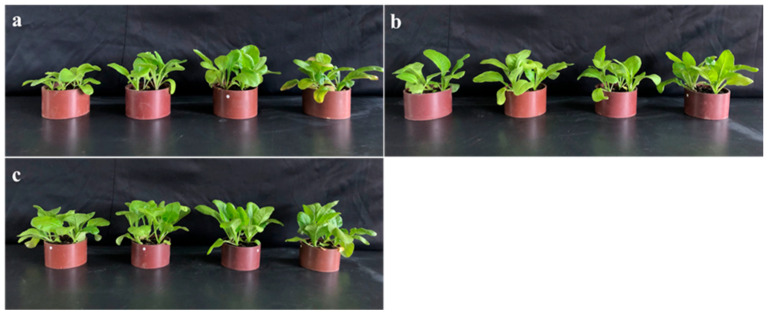
Physical image of vegetable growth (**a**–**c**); the addition of hydrogel is 0%, 2%, and 4%. From left to right, the amount of cadmium added is 500, 50, 5 mg/kg, and 0 mg/kg.

**Figure 8 plants-12-04115-f008:**
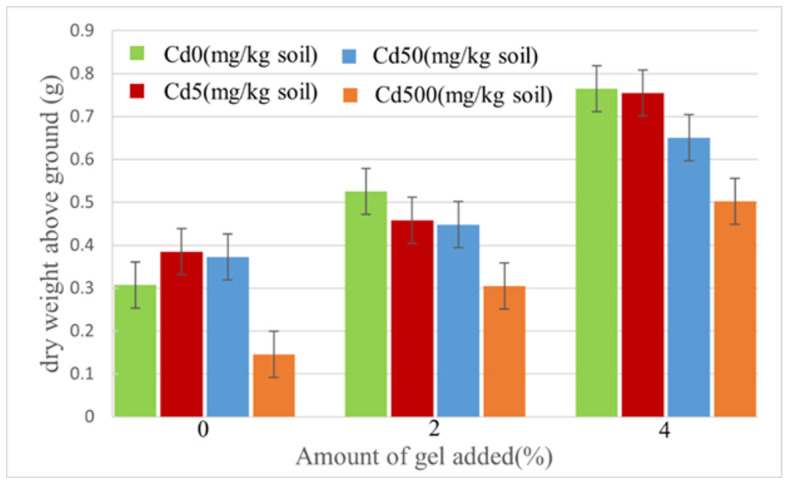
Dry weight of Swiss chard shoots.

**Figure 9 plants-12-04115-f009:**
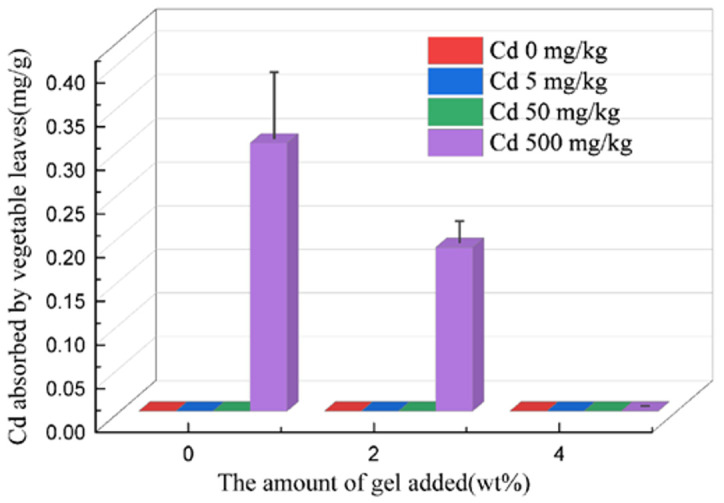
Cadmium absorption by Swiss chard shoots.

**Figure 10 plants-12-04115-f010:**
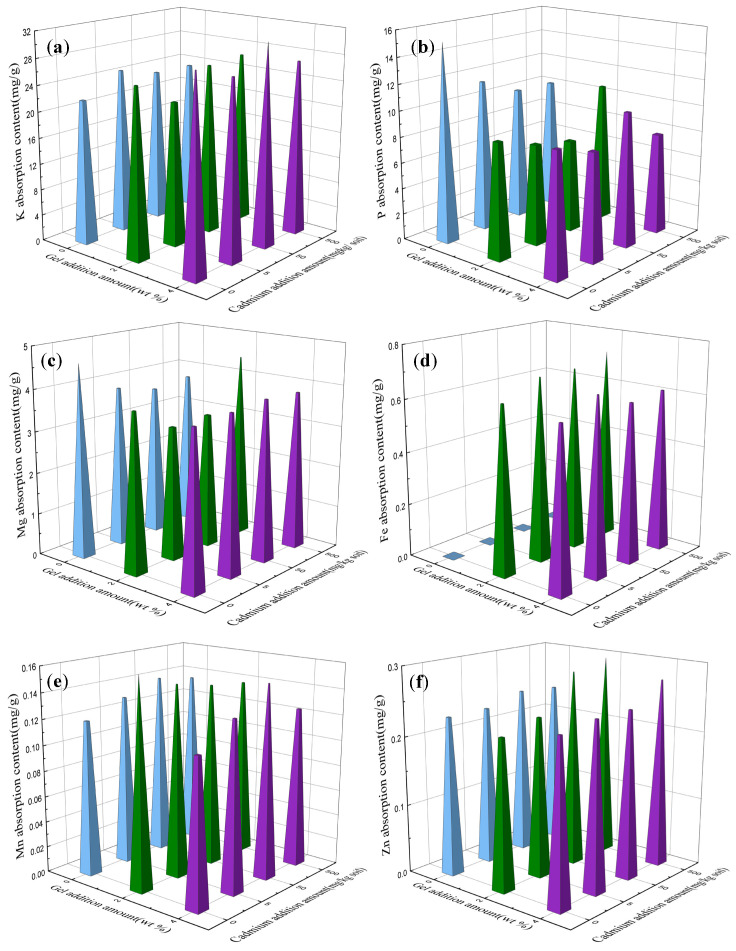
Contents of various elements in vegetable leaves: (**a**) potassium; (**b**) phosphorus; (**c**) magnesium; (**d**) iron; (**e**) manganese; (**f**) zinc; (**g**) copper; (**h**) calcium; (**i**) sodium. Blue: 0 wt.%; Green 2wt.%; Purple 4wt.%-gel content in soil.

**Figure 11 plants-12-04115-f011:**
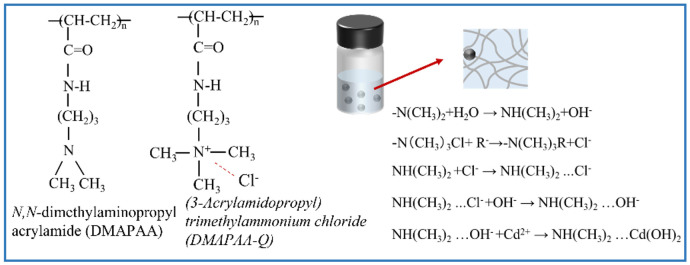
Mechanism diagram of hydrogel adsorption of heavy metals.

**Table 1 plants-12-04115-t001:** Parameters of Langmuir and Freundlich isotherm models for DMAPAA/DMAPAAQ gel at pH = 5.7 and pH = 7.3 for cadmium adsorption capacity.

Isotherm Model	Parameter	Initial pH	
		5.7	7.3
Langmuir	q_max_ (mg/g)	23.7	121
	K_L_ (L/mg)	0.0098	0.0027
	R^2^	0.8537	0.9526
Freundlich	K_F_ (mg/g)	1.55	1.13
	n	2.397	1.492
	R^2^	0.8164	0.9515

**Table 2 plants-12-04115-t002:** DMAPAA/DMAPAAQ hydrogels’ adsorption thermodynamic parameters for cadmium.

T/K	${∆G}^{\theta }/(kJ\cdot {\mathrm{mol}}^{-1})$	${∆H}^{\theta }/(kJ\cdot {\mathrm{mol}}^{-1})$	${∆S}^{\theta }/(J\cdot {\mathrm{mol}}^{-1}\cdot K^{-1})$
298.15	−3.082	4.677	26.025
308.15	−3.342	4.677	26.025

**Table 3 plants-12-04115-t003:** Fit results of Cd^2+^ adsorption kinetics of DMAPAA/DMAPAAQ hydrogels.

Isotherm Model	Parameter	Initial pH	
		5.7	7.3
Pseudo first order	k_1_ (min^−1^)	0.007	0.007
	q_e_ (mg/g)	22.52	32.87
	R^2^	0.967	0.979
Pseudo second order	K_2_ (g/mg/min)	3.818	2.539
	q_e_ (mg/g)	24.76	36.37
	R^2^	0.951	0.939

**Table 4 plants-12-04115-t004:** Synthesis conditions of the DMAPAA/DMAPAAQ hydrogels.

Materials	Component Type	Molar Weight (g/mol)	Concentration (mol/m^3^)	Mass (g)
DMAPAA	Monomer	105.22	500	1.315
DMAPAA-Q	Monomer	206.71	500	2.584
MBAA	Linker	154.17	50	0.193
TEMED	Accelerator	116.21	20	0.058
APS	Initiator	228.19	20	0.114

## Data Availability

All data included in this study are available upon request by contact with the corresponding author.

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
