# Peer review of "Dual Benefits of Hydrogel Remediation of Cadmium-Contaminated Water or Soil and Promotion of Vegetable Growth under Cadmium Stress"

_plants, 2023, doi:10.3390/plants12244115_

Round 1

Reviewer 1 Report

Comments and Suggestions for Authors

The manuscript by Jin et al presents a new hydrogel formula to be applied in cadmium contaminated media. The authors synthetized and attempted to apply the hydrogel in aqueous and soil media at different Cd concentrations and pH values. The work is interesting and could provide valuable information in the field on condition the experiments and the results were properly described and discussed. However, this is not the case. Unfortunately, the manuscript is not properly written in terms of general rules of scientific papers, there are lots of information missing, contains scientifically wrong statements and some English correction needed.

For this reason the paper is unacceptable in the present form. It should be entirely rewritten and consultation with a plant physiologist/plant nutritionist as well as a native English speaker is strongly suggested.

A few examples for the above problems:

Abstract

“121 mg” “0.76 g” This sort of data does not bear information alone. I think it is enough to say that the dry weight of the vegetables was the largest.

Introduction

Capital letter required to start the sentence with. This applies throughout the text

I think that the abbreviations should be explained to the reader: “DMAPAA/DMAPAAQ”

This applies throughout the text – there are many abbreviations without explanation.

Results

2.1. 1st paragraph: Silicon and aluminium are not considered essential for plants to date.

Furthermore, in most soils there are other elements in high concentrations, too. Perhaps, those are not detected by EDS

2.2. The text in 2.2 mainly fits the Discussion. It is not the description of the results but a description of the mechanism of adsorption by the hydrogel. And there is no reference which would be missing if this part had gone to the Discussion.

2.3. I think that the description of the models should go to the Materials and Methods. This applies to other subchapters as well.

Figure 5. Reference to diagram ‘a’ is missing from the caption.

2.74. K, P and Mg are not trace elements but essential macroelements. The concentration of these elements does not reflect the intensity of photosynthesis. Na is not essential element

Discussion

Does not contain a single reference!

Soil is not an ecosystem!

Figure 11. is intended to describe the adsorption mechanism. However, I can see only a not precisely written equations and some intentions on what is happening. I think that it is not a clear description of the process.

Materials and methods

The majority of abbreviations applied here needs explanation!

All instruments should be described: type, manufacturer, city, country.

4.41. The title is not represented here but the Cd concentration

ICP instrument description and sample preparation parameters are missing.

4.5. How was the soil divided? In the pot? If divided why the soil was put into the pots first and then treated? What kind of seeds were used?

4.51 In 4.5 only 4 seeds were reported to be planted /pot. Then, how did you choose 8 plants in each pot?

Conclusions

What about the longevity of the hydrogel? If added to the soil how long does it prevent Cd uptake by plants?

Comments on the Quality of English Language

Some grammatical mistakes can be found in the text. Some spelling errors can be found also.

Reviewer 2 Report

Comments and Suggestions for Authors

This study is very important for soil heavy metal pollution remediation, providing a more reasonable and efficient solution to control soil Cd pollution and ensure the safety of vegetable planting. The experiment design is clear, the data is detailed and the analysis is reasonable. The content is consistent with the requirements of the topic of this publication. It can be accepted after minor revision. Specific questions are as follows:

(1) Fig. 3, Fig.4: No parallel samples, no error bars?

(2) Fig.5: where is (a) in caption?

(3)Conclusion: The conclusion is a summary and extension of the results of this study, without the need to label the literature.

(4) why chose the three concentrations of Cd? Is it consistent with the actual contaminated soil conditions, and can the results be shown to be suitable for practical application? Actual contaminated soil have been tried to test the effect?

(5) why chose the two dosage of gel? The results showed that 4% dose effectively reduced plant Cd absorption and inhibited the decrease of biomass, but the biomass was still reduced (Cd: 500 mg/kg), and the plant Cd absorption had no effect. What is the recommendation of this result for practical application of dose addition?

(6)addition of gel reduces the absorption of Cd in plants, but does not affect the absorption of other metals. FTIP spectrum can be used to test other metals adsorption?

Comments on the Quality of English Language

no comments

Reviewer 3 Report

Comments and Suggestions for Authors

The article 'Dual benefits of hydrogels in water and soil remediation, treatment of cadmium pollutants and promotion of vegetable growth' provides good scientific explanation of metal adsorption by hydrogels and its significance on plant growth. I have no major concern regarding the article. The units shown in the graphical illustrations may be reconsidered to be unified for e.g., fig. 9  mg/g vs mg/kg.

Round 2

Reviewer 1 Report

Comments and Suggestions for Authors

The manuscript by Jin et al has been revised by the authors and it has improved a lot. However, there are several issues that has not been addressed adequately. Some of them are crucial.

I list here only the crucial ones I have found after second evaluation, the rest is in the annotated pdf I attach for the revision.

Abstarct

In the last lines it is written that Cd removes Cd from the soil. In my point of view it is not removed as it still remains in the soil volume but once hydrogel is added to the same volume of soil Cd is adsorbed and transformed. (Please explain if this is not the case) So, I suggest to replace ‘soil’ with ‘soil bioavailable fraction’.

Discussion

In my previous evaluation I have indicated that the discussion comes without any references. In my point of view this is unacceptable as the Discussion means that the findings (the results of the work) is not only explained in free composition but they are compared to those of other authors. If there is no other work describing hydrogels (being a plant biologist I do not know) but there are lots of other publications describing methodologies to decrease cadmium availability in the soil (some of them are mentioned without reference somewhere in the text). For example the efficiency of other methods can be opposed to the applied hydrogel. So these evaluations are missing and without them the work is unacceptable. I have indicated in the annotated version some statements where I think a reference would be important.

Conclusions

The Conclusion here is not acceptable. The text is mostly the repetition of results.

For example it is a good point here to state which is the better model to characterise hydrogels.

But, other specific details are not to be mentioned again. Instead, we need implications in general based on the results of the study. Here the authors must generalize.

Round 3

Reviewer 1 Report

Comments and Suggestions for Authors

The authors have made all suggested corrections so I think the paper can be accepted for publication in Plants.

Comments on the Quality of English Language

I think that the language editor can handle the remaining English language mistakes/errors.